



# Review and syntheses: Turbidity flows – evidence for effects on deep-sea benthic community productivity is ambiguous but the influence on diversity is clearer

Katharine T. Bigham[1, 2], Ashley A. Rowden[1, 2], Daniel Leduc[2], David A. Bowden[2]

[1]School of Biological Sciences, Victoria University of Wellington, Wellington, 6140, New Zealand
[2]National Institute of Water and Atmospheric Research, Wellington, 6021, New Zealand

*Correspondence to*: Katharine T. Bigham (katie.bigham@vuw.ac.nz)

**Abstract.** Turbidity flows – underwater avalanches – are large-scale physical disturbances that are believed to have profound and lasting impacts on benthic communities in the deep sea, with hypothesised effects on both productivity and diversity. In this review we summarize the physical characteristics of turbidity flows and the mechanisms by which they influence deep sea benthic communities, both as an immediate pulse-type disturbance and through longer term press-type impacts. Further, we use data from turbidity flows that occurred hundreds to thousands of years ago as well as three more recent events to assess published hypotheses that turbidity flows affect productivity and diversity. We found, unlike previous reviews, that evidence for changes in productivity in the studies was ambiguous at best, whereas the influence on regional and local diversity was more clear-cut: as had previously been hypothesized turbidity flows decrease local diversity but create mosaics of habitat patches that contribute to increased regional diversity. Studies of more recent turbidity flows provide greater insights into their impacts in the deep sea but without pre-disturbance data the factors that drive patterns in benthic community productivity and diversity, be they physical, chemical, or a combination thereof, still cannot be identified. We propose criteria for data that would be necessary for testing these hypotheses and suggest that studies of Kaikōura Canyon, New Zealand, where an earthquake-triggered turbidity flow occurred in 2016, will present helpful insights into the impacts of turbidity flows on deep-sea benthic communities.

## 1 Introduction

Turbidity flows are a type of large-scale physical disturbance that is prevalent in the deep sea (i.e., at water depths >200 m). They are a component of gravity-driven sediment flows (Nardin et al., 1979) that occur in every ocean basin across the globe (Heezen et al., 1955; Levin et al., 2001; Weaver and Rothwell, 1987). The flows form when submarine landslides disintegrate and mix with seawater (Talling et al., 2014) creating high density parcels of turbid water filled with suspended sediment (Kuenen and Migliorini, 1950) that travel downslope beneath less dense seawater (submarine landslides and turbidity flows are sometimes collectively known as 'mass wasting' or 'mass sediment movement' events). Submarine landslides occur along continental margins even at fairly low slope gradients but are most frequent on steep-sided



30 geomorphic features such as canyon walls, seamount flanks, and ocean trench walls and ridges (Hughes Clarke et al., 1990; Masson et al., 1996; Nardin et al., 1979). Initiation of submarine landslides can be triggered by over-steepening of slopes due to sediment build up, increase in pore pressure, typhoons and atmospheric storms, river flooding and outflow, benthic storms, and earthquakes (Carter et al., 2012; Heezen et al., 1955; Meiburg and Kneller, 2010; Solheim et al., 2005; Talling, 2014; Talling et al., 2013). In situ measurements have recorded turbidity flow velocities greater than 120 cm s$^{-1}$

35 (Khripounoff et al., 2003) and shown flows carrying heavy objects (800 kg) or moving moorings down-canyon (Heerema et al. 2020).





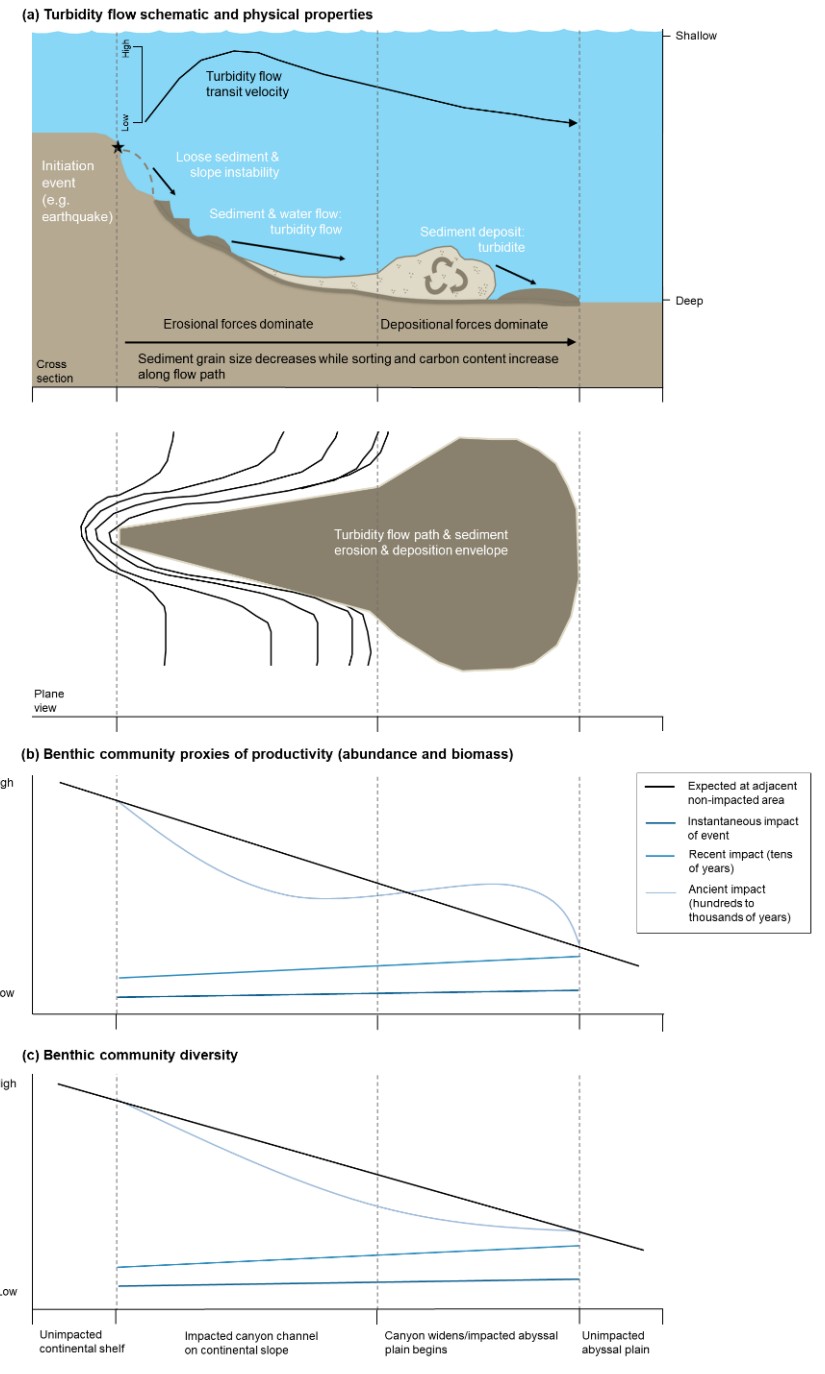

**Figure 1: Generalized schematic of a large, single flow, earthquake triggered turbidity flow within a continental slope canyon with environmental variables at the moment of the event (adapted from NOAA, Talling et al. 2013, Talling et al. 2014, Stetten et al. 2015, Heerema et al. 2020). (b-d) Our hypothesised response of the benthic community productivity (abundance and biomass) and**



**diversity along the turbidity flow path at three time points (the instantaneous moment of the event, tens of years after, and hundreds to thousands of years after).**

The frequency of turbidity flows is highly variable; some can occur annually to decadally (Dennielou et al., 2017; Heezen et al., 1964; Liao et al., 2017; Vangriesheim et al., 2009) while others such as those initiated by earthquakes > 7.0 Mw off the coast of Japan are estimated to occur every 30-900 years (Bao et al., 2018; Yamanaka and Kikuchi, 2004) (Fig. 2). The size and scale of turbidity flows in the deep sea is variable with the spatial scale being dependent on the quantity of source material, distance from continental margins, basin morphology, and bathymetric gradient (Gorsline, 1980).

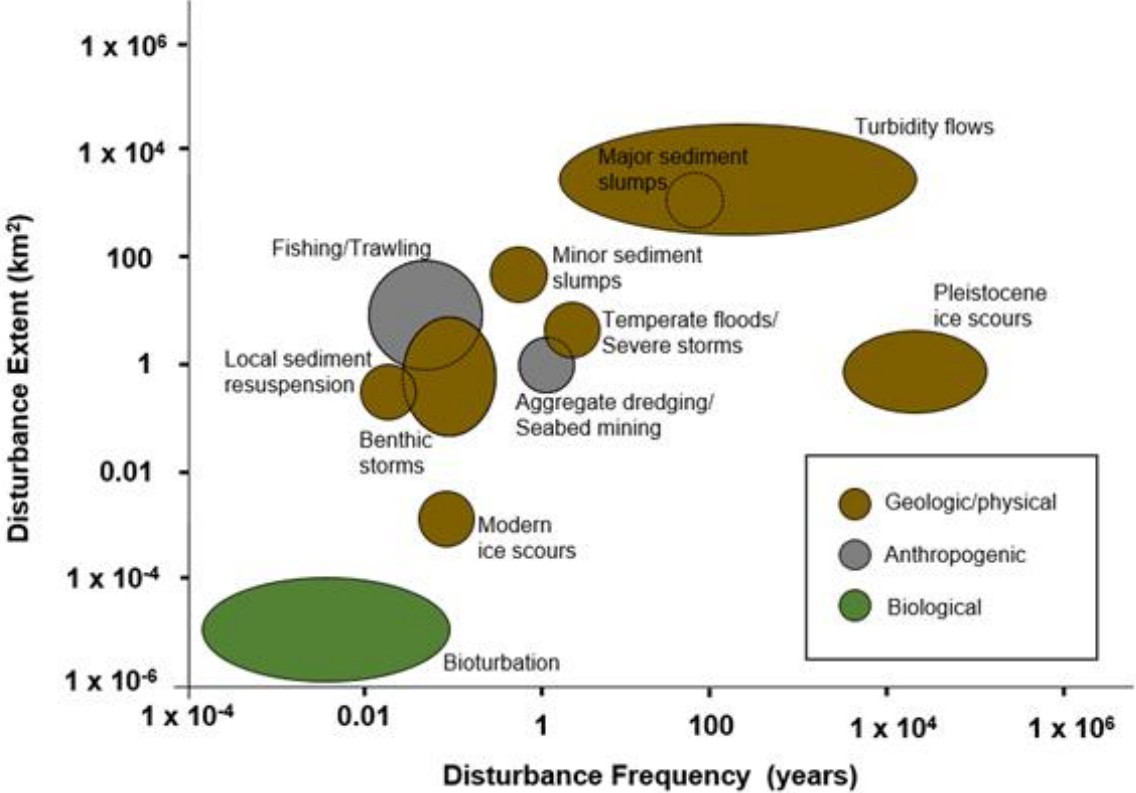

**Figure 2: Graph summarizing the frequency (years) and extent patch size (km²) of physical disturbances in the marine environment. Specific data ellipses are not attributed to particular sources because each is based on our interpretation of figures and data from Hall et al. 1994, Glover et al. 2010, Harris et al. 2014, and references therein.**

Turbidity flows transport massive volumes of sediment and associated organic material from the near shore environment into the deep sea. Ancient turbidity flows (e.g., Grand Banks, and others noted below) have transported an estimated average volume of 0.525 km³ to 185 km³ (Griggs and Kulm, 1970; Heezen and Ewing, 1952; Weaver and Rothwell, 1987) of organic-rich sediment (0.25% - 2.2% organic carbon) (Briggs et al., 1985; Griggs et al., 1969; Huggett, 1987) at scales of tens to hundreds of thousands of km² (see references in Sect. 2.1). At these scales turbidity flows present a major hazard to





human infrastructure such as submarine telecommunication cables and oil and gas platforms (Carter et al., 2014; Heezen et al., 1964; Heezen and Ewing, 1952; Hsu et al., 2008; Hughes Clarke et al., 1990; Solheim et al., 2005) as well as causing

destructive disturbance to benthic faunal communities.

Physical   disturbances, such as turbidity flows, are a structuring factor for biotic communities in all environments (Churchill and Hanson, 1958; Dayton, 1971; Dial and Roughgarden, 1998; Levin and Paine, 1974; Paine, 1979; Raup, 1957; Sousa, 1979; Weaver, 1951; Webb, 1958). In the marine environment disturbances can be caused by natural physical processes such as the battering of intertidal regions by tides and storms (Dayton, 1971; Levin and Paine, 1974; Paine, 1979), biological

processes such as bioturbation of sediment (Hall et al., 1994; Preen, 1996; Reidenauer and Thistle, 1981; Thrush et al., 1991), and anthropogenic impacts such as those arising from bottom trawling (Collie et al., 2000; Lundquist et al., 2010; Thrush et al., 1998). Compared to other physical disturbances experienced by benthic communities, turbidity flows represent a major disturbance (Fig. 2).

Pickett and White (1985) define a disturbance as "…any relatively discrete event in time that disrupts ecosystem,

community, or population structure and changes resources, substrate availability, or the physical environment." By creating patches and freeing limiting resources (space, refuge, nutrients, etc.) disturbances structure ecological succession and increase habitat heterogeneity thus enhancing biodiversity (Hall, 1994; Sousa, 1984, 2001; Willig and Walker, 1999). Disturbances are measured by their frequency (the number of events per unit of time), extent (spatial area of impact), and two components of magnitude: intensity (physical force of the event), and severity (consequence to some component of the

ecological system) (Sousa, 2001). Disturbances have been categorized into two types: pulse (disturbances which have immediate and instantaneous impacts) and press (disturbances that operate over prolonged periods of time) (Bender et al., 1984). These characteristics of a disturbance, along with the functional diversity (based on biological traits such as feeding mode) of the impacted community, influence the resilience of the biotic community to the disturbance, i.e., how resistant it may be   or how quickly it may be able to recover following disturbance (Folke et al., 2004; Holling, 1996; Naeem and

Wright, 2003; Oliver et al., 2015; Walker et al., 2004).

For seabed or benthic communities, as already noted above, the effect of a disturbance varies within a community depending on the characteristics of the disturbance as well as the biological characteristics or traits of the impacted organisms. For example, the more mobile the organism the greater its likelihood of escaping the disturbance altogether, either by being able to burrow below the area impacted by the event (in the case of small organisms) or leave the area altogether (in the case of

larger megafaunal organisms) (Crandall et al., 2003). Additionally, small mobile organisms can sometimes burrow upwards when disturbances bury them rather than being smothered (Maurer et al., 1986; Nichols et al., 1978; Tiano et al., 2020). It can be harder to determine the impact of disturbances to mobile organisms such as demersal fish compared to sessile organisms, because they may not be killed outright by the event. Newly exposed or dead benthic fauna resulting from turbidity flows can provide an immediate and concentrated source of food for fish (Okey, 1997). However, after such short-

term benefits are exploited, because disturbances mostly eliminate or create shortages of vital resources such as food or cover for mobile organisms, populations tend to decline until these resources have regenerated (Sousa, 1984).



While we know that turbidity flows can have damaging impacts on seabed communities in the deep sea (see Sect. 2.2), we do not understand clearly how benthic communities respond to these catastrophic events or how patterns of benthic productivity and diversity are influenced by them. As noted by Glover et al. (2010), turbidity flows have historically been studied via palaeontological proxies due to their size and the time scales at which they occur (Fig. 2). These proxies are identifiable by their characteristic deposition of graded sediment known as turbidites (Kuenen and Migliorini, 1950).

This review is timely because the previous review on the topic was limited to studies of ancient turbidites (Young et al., 2001) and a growing body of studies has been generated on more recent turbidity flows. This review was further prompted by the 2016 Kaikōura Earthquake (New Zealand) and subsequent turbidity flow which presents an exceptional opportunity to advance our understanding of turbidity flows' impacts on deep-sea benthic community. Here we evaluate published data on turbidity flows to assess the influence of this type of disturbance on: (1) benthic community productivity; (2) local and regional diversity of benthic communities; and (3) consider further research to address the gaps in our understanding of how turbidity flows impact benthic communities in the deep sea.

## 2 Turbidity Flows

### 2.1 Examples of deep-sea turbidity flows

The classic case study of a turbidity flow is that described by Heezen and Ewing (1952) following the Grand Banks Earthquake in November 1929. Following the 7.2 (Mw) earthquake on the continental slope south of Newfoundland (Canada), submarine telegraph cables extending along the continental slope at different depths were broken in an orderly progression, whereas there was no damage to cables on the continental shelf. Heezen and Ewing (1952) argued that the earthquake caused a slump on the continental slope that incorporated water to form a fast-moving turbidity flow that travelled 1100 km from its source. The sediment or turbidite deposited by this turbidity flow on the Sohm Abyssal Plain to the south of the cable breaks has since been estimated to cover an area of 160,000 km$^2$ with a volume of 185 km$^3$ (Masson et al., 1996).

Three well-studied but ancient turbidity flow sites were reviewed by Young et al. (2001); those that have occurred in the Cascadia Channel, the Venezuela Basin, and the Madeira Abyssal Plain. The Cascadia Channel, adjacent to the states of Oregon and Washington on the west coast of the United States of America (USA), has evidence of multiple turbidites originating from the Columbia River drainage that collectively extend at least 650 km along the channel axis and are estimated to have transported on average 0.525 km$^3$ of sediment (Griggs et al., 1969; Griggs and Kulm, 1970). The last recorded turbidity flow occurred in the Cascadia Channel around 6600 years ago (Nelson et al., 1968). The Venezuela Abyssal Plain in the Caribbean Sea received regular turbidity flows of organic-rich terrestrial materials from the Magdalena Fan, the last one having occurred around 2000 years ago (Young and Richardson, 1998). The Madeira Abyssal Plain off northwest Africa consists of multiple turbidite layers (up to 5 m thick) interspersed with layers of pelagic clays (centimetre to decimetre thick). The Madeira Abyssal Plain turbidites have estimated volumes ranging from 4.5 km$^3$ to 126 km$^3$ and



cover an area of up to 80,000 km² (Weaver and Rothwell, 1987) the most recent having occurred around 930 years ago
125  (Thomson and Weaver, 1994). More recent turbidity flows, which have not previously been reviewed, include the 1999
storm-triggered turbidity flow in Cap Breton Canyon, France (Anschutz et al., 2002; Hess et al., 2005), periodic turbidity
flows in the Congo Channel off southwest Africa (Khripounoff et al., 2003; Vangriesheim et al., 2009), and a turbidity flow
off the coast of Japan triggered by the 9.0 (Mw) 2011 Tōhoku Earthquake that transported an estimated 0.2 km³ of sediment
(Kioka et al., 2019) (Fig. 3 and Table 1).

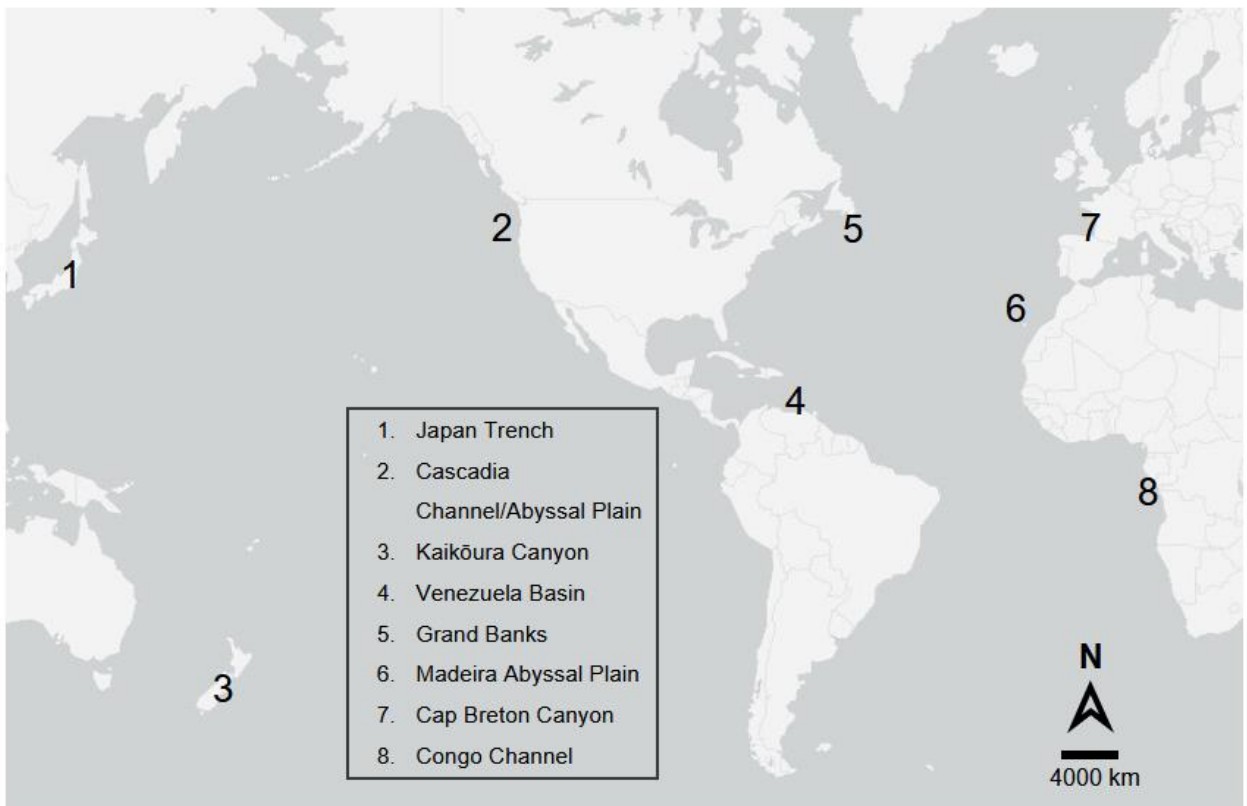

130

**Figure 3: Map showing the approximate location of the turbidity flows reviewed in this paper. See Table 1 for additional details.**





| Turbidite site | Latitude | Longitude | Depth (m) | Trigger | Year of event | Age of turbidite at time of study (years) | Volume (km³) | Fauna size class studied | Studies used in Fig. 4 and Fig. 5 | References for metadata |
|---|---|---|---|---|---|---|---|---|---|---|
| Cascadia Channel/Abyssal Plain, NE Pacific | 44°40'N | 127°20'W | 2900 | Unknown | Unknown | Approx. 6600 | 0.525 | Macrofauna, Megafauna | Griggs et al. 1969*; Carey 1981; Pearcy et al. 1982 | Nelson et al. 1968; Griggs et al. 1970 |
| Venezuela Basin, Caribbean Sea | 13°45'N | 67°45'W | 5050 | Unknown | Unknown | Approx. 2000 | Unknown | Meiofauna, Macrofauna, Megafauna | Tietjen 1984; Richardson et al. 1985*; Woods and Tietjen 1985*; Briggs et al. 1996*; Lambshead et al. 2001 | Young and Richardson 1998 |
| Maderia Abyssal Plain, Mid-Atlantic | 31°N | 21°W | 4950 | Unknown | Unknown | Approx. 930 | 4.5–126 | Meiofauna, Macrofauna, Megafauna | Thurston et al. 1994*; Lambshead et al. 1995; Gooday 1996; Glover et al. 2001; Lambshead et al. 2001 | Weaver and Rothwell 1987 |
| Grand Banks, NW Atlantic | 44°N | 55°W | 220–4800 | Earthquake | 1929 | 59 | 185 | NA | NA | Heezen and Ewing 1952; Mayer et al. 1988; Masson et al. 1996 |
| Cap Breton Canyon, NE Atlantic | 43°37'N | 01°43'W | 251–1478 | Storm | 1999 | 18 | Unknown | Meiofauna, Macrofauna | Hess 2005; Hess and Jorrison 2009**; Frutos and Sorbe 2017 | Anschutz et al. 2002; Hess 2005 |
| Congo Channel, SE Atlantic | 05°43'S | 08°27'E | 3964 - 4960 | River Flooding | 2001, 2004 | < 1 year, Unknown | Unknown | Meiofauna, Macrofauna | Galéron et al. 2009; Van Gaever et al. 2009; Olu et al. 2017 | Khripounouff et al. 2003; Vangriesheim et al. 2009 |
| Japan Trench, NW Pacific | 38 °N | 142°E | 300–7556 | Earthquake | 2011 | 8 | 0.2 | Meiofauna | Kitahashi et al. 2014; Kitahashi et al. 2016; Nomaki et al. 2016; Kitahashi et al. 2018; Tsujimoto et al. 2020 | Oguri et al. 2013; Oguri et al. 2016; Kioka et al. 2019 |
| Kaikōura Canyon, SW Pacific | 42 °S | 173°E | 900–1200 | Earthquake | 2016 | 2 | 0.9 | NA | NA | De Leo et al. 2010; Mountjoy et al. 2018 |

**Table 1: Metadata for turbidites and studies discussed and analysed in this review. *Study was reviewed in Young et al. (2001). **Includes data from Hess (2005)**

## 2.2 Initial impacts of turbidity flows on benthic communities

Turbidity flows act as both press- and pulse-type disturbance with erosional and depositional forces (Harris, 2014). The immediate pulse-type impact is mass mortality, either due to dislodgment by erosional forces or burial by deposition. Organisms caught up in the erosional forces become part of the material transported to deeper depths, and it is generally assumed that even if they are not killed outright, they would be unable to establish themselves in depositional environments that may be tens to hundreds of kilometres away and at deeper water depths (Griggs et al., 1969). However, some studies have found evidence of range extension of shallow water taxa in areas with regular sediment mass movements that might be the result of acclimation of transported fauna (Kawagucci et al., 2012; Rathburn et al., 2009; Tsujimoto et al., 2020).

It has been suggested that the impacts of burial in areas of deposition may be greater for deep-sea fauna than the erosional forces (Miller et al., 2002). Whether deep-sea organisms will survive the deposition of a turbidite depends on depth of sediment deposition, the type of material being deposited compared to the sediment already there, and the impacted organisms (Young and Richardson, 1998). In shallow water environments where sedimentation rates are higher and physical disturbances more frequent, studies have shown some organisms are able to survive burials between 30 and 50 cm, with the more mobile fauna able to migrate up through the deposited material to the sediment-water interface (Maurer et al., 1986; Nichols et al., 1978). Survival rates are higher if the deposited material is similar to the underlying sediment (Kranz, 1974). Organisms tend to be adapted for a particular substrate type and unlike their shallow water counterparts, deep-sea organisms have evolved to survive very low sedimentation rates; typically 0.1-2.9 cm kyr$^{-1}$ for the abyssal plain (Stordal et al., 1985; Weaver and Rothwell, 1987). Because of this general lack of adaptation to deep burial, it has been predicted that sessile deep-sea fauna in depositional zones of the turbidity flow would be killed by as little as a few millimetres to ten centimetres of sediment (Jumars, 1981). The more mobile the organism the better its chance of avoiding being buried (Young and Richardson, 1998).



The unconsolidated nature and sometimes quite fine grain size of the sediment transported by turbidity flows means that the impacts of the mass wasting are often spread over a large area (Lambshead et al., 2001). The clogging impact of increased turbidity, especially to filter and suspension feeders, may persist for long periods following the triggering event. Turbid waters were observed in Sagami Bay for up to three hours following the 5.4 (Mw) Off-Izu Peninsula Earthquake in April

2006 off Japan (Kasaya et al., 2009). This turbid water prolongs the impact of turbidity flows triggered by earthquakes as aftershocks can cause the recently transported fine grain sediments to be resuspended into nepheloid layers (layers that contain significant amounts of suspended sediment). After the 2011 Tōhoku Earthquake off Japan, increased turbidity was observed at multiple sites, up to three months after the initial earthquake (Kawagucci et al., 2012). These nepheloid layers ranged between 30 and 50 m above the seafloor (Oguri et al., 2013). Persistent turbidity could delay an organism's ability to

settle the disturbed patches of the seafloor or even, in the case of aftershocks, cause additional mortalities of colonising fauna. A comparison of the ratio of detritus to filter feeders in the Congo Channel found a higher proportion of detritus feeders in the distributary system than in adjacent areas, which may reflect the possible deleterious impacts of turbidity flows on filter feeders as opposed to the potential benefit to detritus feeders (Heezen et al., 1964)

Another way in which turbidity flows act as press-type disturbances and may delay faunal community response to the newly

available sediment deposits is the creation of anoxic and hypoxic conditions. The introduction of large volumes of organic matter, either from organic-rich coastal sediments or organisms caught up in the mass sediment movement, can lead to anoxic conditions developing as bacteria break down the newly settled and buried organic matter. The presence of these near-surface reducing zones has been observed as layers of thin, iron-rich crust in the Venezuela Basin turbidites (Briggs et al., 1985). These anoxic or hypoxic conditions restrict organisms that rely on aerobic respiration and as a result may delay

recruitment of benthic fauna as colonisation cannot begin until oxygen is again present in the bottom water (Froelich et al., 1979). Low oxygen persistence is a function of the volume and organic content of the turbidite, speed and direction of the benthic boundary layer currents, and diagenetic processes in the sediment (Sholkovitz and Soutar, 1975).

Turbidity flows can create completely new habitats in the deep sea not only by removing existing faunal communities but also by uncovering or creating new resources. For example, chemosynthetic communities; unique assemblages of organisms

that are fuelled by the chemosynthesis of reduced chemical compounds rather than photosynthetic detritus (Sibuet and Olu, 1998), will occur where there is enough organic material to support reducing conditions (Gooday et al., 1990). Turbidity flows initiate the development of chemosynthetic communities both through the burial of large volumes of organic material and through exposure of methane bearing sediments by erosion (Rathburn et al., 2009). Chemosynthetic communities have been observed at the Laurentian Fan in the path of the Grand Banks turbidity flow, in Monterey Canyon slide scarps, and at

the Congo deep-sea fan (Embley et al., 1990; Mayer et al., 1988; Savoye et al., 2000). In the Laurentian Fan the chemosynthetic communities were associated with gravel exposed by the Grand Banks turbidity flow that allowed methane rich fluids to percolate to the surface (Mayer et al., 1988). The chemosynthetic communities in Monterey Canyon were also associated with the erosional environments of ~100-year-old slump scars, and it was predicted that these communities would persist as long as the methane was available (Paull et al., 2010). At the Congo deep-sea fan, the chemosynthetic communities





are associated with turbidite deposits of terrigenous material from the Congo river basin that was transported through the Congo Channel (Pruski et al., 2017; Stetten et al., 2015). The communities, as dense as  cold seep communities in typical continental margin settings (Olu et al. 2017), consist of mobile seep endemic and vagrant species that are adapted to take advantage of the abundant resources, while tolerating periods of high stress from burial and erosion (Sen et al., 2017).

## 3 Response of benthic communities to turbidity flows

### 3.1 Influence on productivity

Prior to the introduction of the concept of turbidity flows, it was widely accepted that all non-chemosynthetic fauna in the abyss (i.e., generally 2000 m or 3000 m to 6000 m) were dependent on a steady fall of detritus from the upper layers of the ocean, either via marine snow or through the rafting of terrestrial matter (Marshall, 1954). A positive correlation between increased organic flux from surface waters to the benthos and greater biomass, larger body size, and higher bioturbation rates
has been observed (Jahnke and Jackson, 1992; Smith, 1992; Trauthl et al., 1997), but these rates of organic flux are well below the rates associated with turbidity flows. Following a 6.8 (Mw) magnitude earthquake in Venezuela in July 1997, the measured carbon flux to the Caraico Basin (1,400 m water depth) was 30 times higher for the two-week period in which the earthquake occurred than the preceding two-weeks (Thunell et al., 1999). Carbon flux measured in situ from a turbidity flow in the Congo Canyon in 2001 was 100 times higher than normal (Khripounoff et al., 2003). In 1955, Heezen, Ewing, and
Menzies proposed that turbidity flows may act as an additional vector for the transport of near-shore or terrestrial materials to the deep sea. They hypothesised that there would be a "high correlation between nutrient rich turbidity current areas and a high standing crop of abyssal animals" (Heezen et al., 1955). They tested their hypothesis at the Congo Canyon where the presence of turbidity flows was again determined from submarine cable breaks. Contrary to normal expectation, they found that the abundance of fauna in biological trawl samples did not decrease with depth along the presumed turbidity flow path
(depth range sampled: 1635 - 2137 m), which they interpreted as a positive influence of the turbidity flow on productivity (Heezen et al., 1964). Despite the suggestive nature of these data from the Congo there was little other evidence at the time and Heezen et al. (1964) themselves acknowledged that more information was necessary to test their central hypothesis that turbidity flows provide a significant carbon subsidy to the abyss that is reflected in increased benthic productivity (e.g., evidenced by increased faunal abundance and/or biomass) in regions most affected by the deposition of turbidites.
Subsequently, changes in sediment characteristics, increases in overall total organic carbon, unique faunal lebensspuren (tracks, burrows and other signs of life and activity on the seafloor, including bioturbation) and faunal assemblages were detected between turbidites and pelagic sediments even thousands of years after the turbidites had been deposited (Huggett, 1987), but Heezen et al.'s hypothesis on benthic faunal productivity remained largely untested.

Young et al. (2001) reviewed several studies from abyssal turbidites (Cascadia Channel-Abyssal Plain, Venezuela
Basin, and Madeira Abyssal Plain) to formally evaluate Heezen et al.'s (1955) hypothesis. They concluded that the data from the studies they examined did not support the hypothesis. However, the studies cited by Young et al. (2001) are not as





conclusive as they were interpreted to be. The Cascadia Channel-Abyssal Plain and Venezuela Basin studies sampled three main sedimentary regimes: turbidite, pelagic, and hemipelagic. Hemipelagic sediments have higher biogenic and terrigenous material than pelagic sediments as they are found on continental slopes beneath highly productive surface waters. At the
Madeira Abyssal Plain the study sites also included an abyssal plain below more productive surface waters, in this case seasonally eutrophic waters, an oligotrophic turbidite, and oligotrophic non-turbidite abyssal plain. Young et al. (2001) focused on the difference between the faunal abundances and biomass in the turbidites and those in the hemipelagic or eutrophic sediments, rather than considering the nearby pelagic sediments.

In the Cascadia Channel-Abyssal Plain, the abundance (1011 animals/m$^2$) and biomass (2.2 wet wt. g/m$^2$) of macrofauna at the turbidite site was consistently higher than at the two nearby pelagic sediment sites (abundance: 330 and 154 animals/m$^2$, biomass: 1.82 and 0.98 wet wt. g/m$^2$), and comparable to the hemipelagic abundance (1170 animals/m$^2$) (Griggs et al., 1969). The four-times-greater faunal abundance in the turbidites compared to pelagic sediments, and comparable abundance to the shallower and highly productive hemipelagic sediments, led Griggs et al. (1969) to support Heezen et al.'s hypothesis.
However, because Young et al. (2001) only considered the hemipelagic biomass (5.57 g/m$^2$) in their comparison, they concluded that these data did not support the hypothesis. Similarly, Young et al. (2001) only note that the biomass was largest for all fauna class sizes in the hemipelagic sediments from the Venezuela Basin studies. However, when comparing the turbidite and pelagic sediments from the original studies the biomass for all faunal size classes was either higher in the turbidite, or there was no significant difference between the two sites (Tietjen, 1984; Woods and Tietjen, 1985; Briggs et al.,
1996). The abundance of meiofauna (nematodes) was also highest in the hemipelagic but there was no significant difference in abundance between the pelagic and turbidite site (Tietjen, 1984). No significant difference in abundance for macrofauna or megafauna was detected between any of the sedimentary regimes (Richardson et al., 1985; Richardson and Young, 1987). A later study found differences in megafaunal abundance between the sites; with the pelagic site having the highest abundances, then the hemipelagic site, and lastly the turbidite site (Briggs et al. 1996). The authors of these original studies
in the Venezuela Basin all attributed the observed variation in abundance and biomass to varying availability of phytodetritus and terrigenous material at the sites (Briggs et al., 1996; Richardson et al., 1985; Richardson and Young, 1987). Thus, the evidence from the original study in the Cascadia Channel – Abyssal Plain support Heezen et al.'s hypothesis, while the studies from the Venezuela Basin are somewhat equivocal. That is, there is evidence that the biomass but not the abundance of all faunal size classes is higher in turbidites than in nearby pelagic sediments.
Thurston et al. (1994) observed that megafaunal abundance and biomass were lower at the Madeira Abyssal Plain (turbidite) compared to the Porcupine Abyssal Plain (non-turbidite) in the northeastern Atlantic. Thurston et al. (1994) attributed these differences between the sites to variation in surface productivity, supplying phytodetritus to the Porcupine Abyssal Plain and not the Madeira Abyssal Plain. In this case Young et al. (2001) noted that sediment trap data showed similar overall fluxes of suspended material to the seafloor at both sites (Honjo and Manganini, 1993; Newton et al., 1994) and therefore, argued that
the turbidite and not phytodetritus was the driving factor of variation observed between megafauna at the two sites.



However, a subsequent study comparing foraminiferal communities, as a proxy for the meiofaunal communities, at the two sites showed that not only was there a higher amount of phytodetritus at the Porcupine Abyssal Plain but there was also foraminiferal communities uniquely suited to exploiting phytodetrital aggregates (Gooday, 1996). Similarly, the dominant megafauna seen at the Porcupine Abyssal Plain were "vacuum cleaner" holothurians, which are well suited to utilizing

phytodetritus aggregates (Thurston et al., 1994). Further, each of these studies included third sites which are close to the Madeira Abyssal Plain and like it in terms of suspended material flux but have not been impacted by turbidites. For both megafauna and foraminifera (meiofauna), there was no significant difference in abundance or biomass between the Madeira Abyssal Plain turbidite site and the additional non-turbidite sites, but abundance and biomass was higher in the Porcupine Abyssal Plain than all other sites (Gooday, 1996; Thurston et al., 1994). A later study by Thurston et al. (1998) specifically

accounted for the turbidite's potential influence on megafauna invertebrates in the NE Atlantic. They selected two sites from the Madeira Abyssal Plain that had not been impacted by turbidites and found that while there were significant differences in abundance and biomass between these two non-turbidite sites, the general trends between the sites (low abundance, low biomass, dominance of non-detritivore taxa) compared to the Porcupine Abyssal Plain non-turbidite site indicated a regional uniformity that supported the previous finding that the Madeira Abyssal Plain turbidite site was similar to other non-turbidite

sites (Thurston et al., 1994, 1998). This conclusion was further supported by a study examining polychaete abundance in the northeast Atlantic (Glover et al., 2001). Thus, evidence from the NE Atlantic does not support Heezen et al.'s (1955) hypothesis that benthic productivity in the abyss is affected by carbon delivered by turbidity flows, albeit 930 years after the deposition of turbidites on the Madeira Abyssal Plain.

Samples of the macrofauna and meiofauna communities from throughout the flow path of turbidity flows in the Congo

Channel and a nearby control site, have been examined by multiple studies that occurred after the review by Young et al. (2001). The study sites include the channel floor which is regularly disturbed by turbidity flows, a levee site that is only impacted by turbidity flows that are large enough to spill over the canyon walls (Savoye et al., 2009) (which occurred during a March 2001 turbidity flow (Khripounoff et al., 2003) but not during a January 2003 event (Vangriesheim et al., 2009)), and multiple sites on the terminal fan that is formed by the periodic deposition of turbidity flows (Savoye et al., 2009). The

multiple sites at the fan are from five lobes which are differently impacted by the turbidity flows, including one 'abandoned' lobe no longer receiving deposited sediment (Dennielou et al., 2017; Sen et al., 2017). The control site is roughly the same water depth (4000 m) and 150 km south of the channel and levee site (Van Gaever et al., 2009; Galéron et al., 2009). Total organic carbon content on the fan ranged between 3.3-3.7% (Stetten et al., 2015) compared to 1.2% at the Congo Channel (Galéron et al., 2009), which suggests that organic matter food availability is largely driven by horizontal downslope

transport processes and not vertical fluxes from the surface. The density of macrofauna at the levee site was lower than a continental margin control site at similar depths (Galéron et al., 2009). Similarly, meiofauna at the channel, the levee, and the fan site closest to the channel (with the highest turbidity flow intensity), also had lower density than the continental margin control site (Van Gaever et al., 2009). A possible reason for these lower faunal densities is that the organic material carried by turbidity flows in the Congo Channel is degraded compared to the pelagic material the continental margin control





site receives (Treignier et al., 2006), which raises the question of whether or not the material transported by turbidity flows is of sufficient quality for the benthic fauna to benefit as Heezen et al. (1955) predicted . Macrofauna at the four sites in the active sediment deposition region of the fan had higher densities than the abandoned lobe reference site, and the levee and continental margin control site sampled by Galéron et al. (Galéron et al., 2009; Olu et al., 2017). The lower macrofaunal densities at the channel and levee, compared to the fan, may be due to the physical disturbance and/or lower organic carbon

content caused by the turbidity flows. Of the fan sites sampled by Olu et al. (2017), the fan site with the highest turbidity flow intensity had the lowest macrofaunal densities. The low meiofaunal densities observed at this same fan site by Van Gaever et al. (2009) were attributed to periodic sediment burial preventing the full development of the community. Further, Galéron et al. (2009) collected data from the levee before and after the March 2001 turbidity flow, and although they observed an increase in density of macrofauna over the course of their sampling they did not attribute this change to the

turbidity flow's influence. The levee and control site also received an increased flux of organic material of pelagic origins during the study period. Since the continental margin control site saw a greater increase in macrofauna density than the levee site during the study period, the authors proposed that rather than positively influencing the community at the levee site via increased food, the disturbance from the turbidity flow may have instead delayed the macrofauna's ability to respond to the pelagic influx (Galéron et al., 2009). Thus, overall the Congo Channel studies provide further equivocal evidence for Heezen

et al. (1955)'s hypothesis. However, they suggest that the location of the faunal community along the turbidity flow path is critical to whether or not the impact is positive or negative.

Three studies evaluating the impact of turbidity flows on the meiofauna and prokaryote communities following the turbidity flow triggered by the 9.0 (Mw) 2011 Tōhoku Earthquake found that in general faunal density was no different after the event, but that vertical distribution in the sediments trended deeper. Kitahashi et al. (2014, 2016, 2018) conducted studies of

the impact on meiofauna along a north (123 – 5604 m water depth) and south line (150 – 3960 m water depth) in the Japan Trench off the coast of Sanriku. These data, collected 4.5 months and 1.5 years after the turbidity flow, were compared to data collected 24-30 years before the turbidity flow along the same northern line (Shirayama and Kojima, 1994). They found that the overall meiofaunal densities had not changed but meiofaunal vertical distributions did change; for up to a year following the event, meiofauna peaks were seen in the subsurface rather than the surface sediments, as was the case in the

original samples (Kitahashi et al., 2014; Shirayama and Kojima, 1994). A similar subsurface peak was observed for foraminifera (meiofauna) at nearby sites (3250 – 3585 m water depth) following the 2011 turbidity flow (Tsujimoto et al., 2020). These anomalous peaks were attributed to an increase in available carbon, for reasons similar to Heezen et al.'s hypothesis, but other explanatory variables such as dissolved oxygen could not be ruled out (Kitahashi et al., 2014, 2016, 2018).

Subsurface peaks in the vertical distribution of meiofauna and macrofauna were also observed in the Congo Channel (Van Gaever et al., 2009; Galéron et al., 2009) and meiofauna in Cap Breton Canyon (Hess et al., 2005). In the Congo Channel the peak for macrofauna was attributed to either the distribution of organic material or the periodic burial by turbidity flows, which favours living deeper as a strategy to avoid disturbance (Galéron et al., 2009). At Cap Breton, the peak was, again,





attributed to the increased carbon but other explanatory factors such as the oxygen content, chemical factors, or grain size

could not be ruled out (Anschutz et al., 2002; Hess et al., 2005; Hess and Jorissen, 2009).

Nomaki et al. (2016) also found a similar anomalous vertical distribution compared to general deep-sea trends in prokaryote and meiofauna communities one year after the 2011 turbidity flow off the coast of Tōhoku (310 – 880 m water depth), near the epicentre of the earthquake. However, unlike Kitahashi et al. (2014, 2016) and the Cap Breton studies, Nomaki et al. (2016) also collected data for chemical and physical properties of the sediment to address what factors were driving the

anomalous distributions. They found that for prokaryotes, peaks correlated with sediment grain size and organic carbon availability, whereas for copepods vertical distribution was correlated with dissolved oxygen content, and for nematodes vertical distribution was correlated with ammonium concentrations (Nomaki et al., 2016). Overall, the studies following the turbidity flows off Japan and in the Cap Breton Canyon, France provide further equivocal evidence for Heezen et al. (1955)'s productivity hypothesis (Fig. 4). Additionally, they provide some insights into what factors (grain size, carbon

content, and chemical peaks) might be driving the communities' responses to turbidity flows. Nonetheless, the studies lack sufficient pre-disturbance and/or environmental data to draw any definitive conclusions.

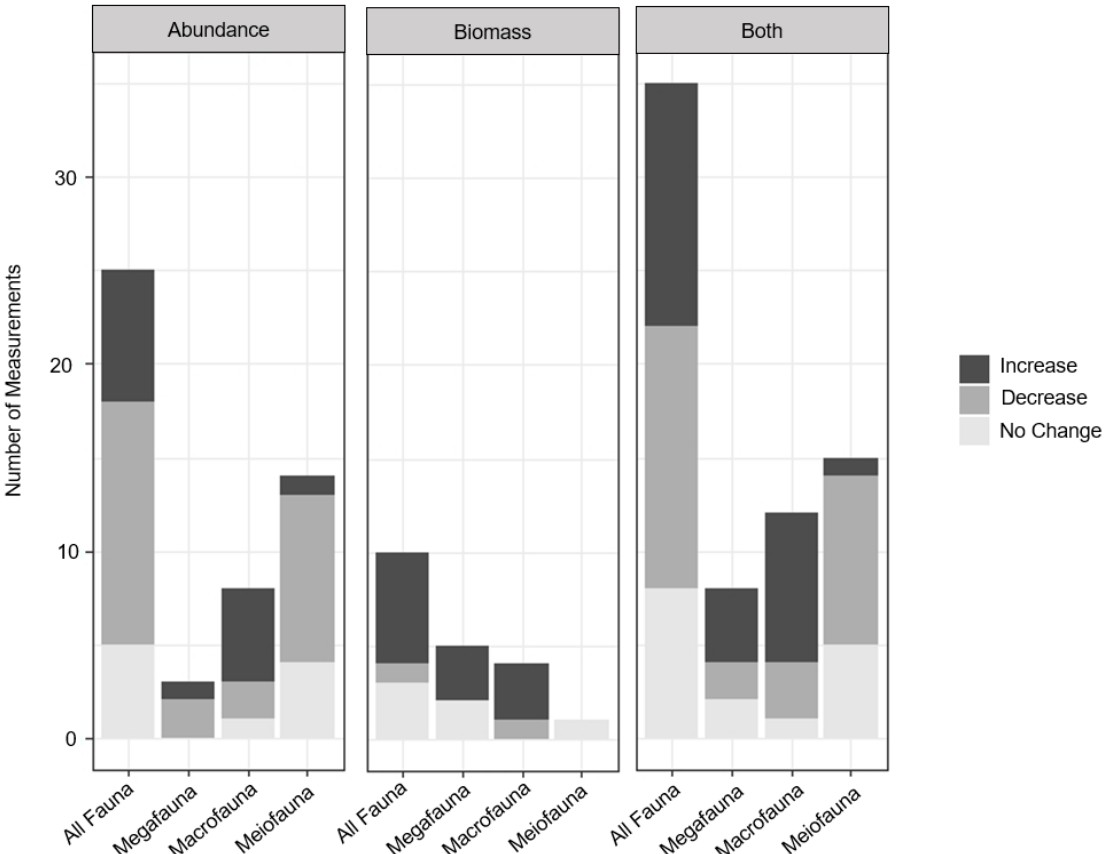

Figure 4: Graphs of the impacts of turbidity flows on    the biomass and abundance (proxies of productivity) of benthic communities in three size classes (mega-, macro-, and meiofauna) and a summary group of all size classes ("All Fauna"). The





**right-hand side graph ("Both") combines all biomass and abundance measurements. Increase, decrease, or no change in abundance or biomass in turbidite-affected areas was assessed by comparing individual measurements of these productivity metrics at different sites in the studies listed in Table 1.**

## 3.2 Influence on diversity

Although Young et al. (2001) dismissed the productivity-based hypothesis of Heezen et al. (1955), after their review of a
small selection of the studies discussed above, they did conclude that turbidity flows likely have an impact on deep-sea faunal diversity as proposed by Angel and Rice (1996). Young et al. (2001) supported the hypothesis that while the initial impacts of turbidity flows can cause local mortalities either by erosional or depositional forces (Fig. 1c) and therefore negatively impact local diversity, overall they contribute to the high regional species richness of deep-sea benthic communities by creating mosaics of habitats in time and space (Angel & Rice 1996).

In general, locally depressed diversity and increased dominance has been observed in turbidites, a pattern typical of disturbed regimes throughout marine environments (Aller, 1997; Glover et al., 2001; Okey, 1997; Paterson and Lambshead, 1995). These general trends of lower local diversity at turbidites as predicted by Angel and Rice (1996) have been found in a number of turbidity flow studies (Briggs et al., 1996; Frutos and Sorbe, 2017; Van Gaever et al., 2009; Glover et al., 2001; Hess and Jorissen, 2009; Kitahashi et al., 2016; Lambshead et al., 2001; Olu et al., 2017; Tsujimoto et al., 2020). While at a
regional scale these same studies and others have noted that the turbidites host unique communities of species not seen at other sites in the region and therefore increase the diversity of the region as a whole (Briggs et al., 1996; Frutos and Sorbe, 2017; Van Gaever et al., 2009; Glover et al., 2001; Hess and Jorissen, 2009; Olu et al., 2017; Tietjen, 1984) and supporting Angel and Rice's (1996) hypothesis (Fig. 5).





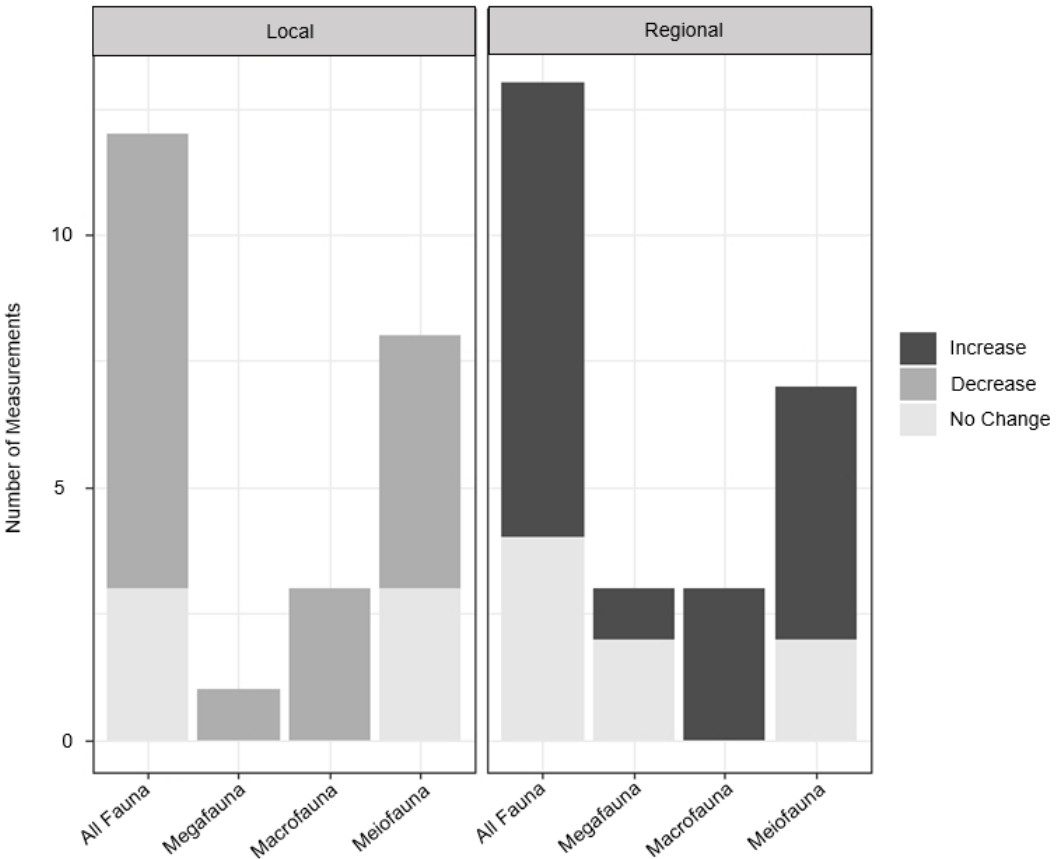

**Figure 5: Graphs of the impact of turbidity flows on diversity in three size classes (mega-, macro-, and meiofauna). For local diversity, the increase, decrease, or no change in species richness or other diversity indices was assessed from individual measurements of these metrics made in the studies listed in Table 1. For regional diversity, the assessment was made based on "unique community" as noted by the authors of these studies, or our interpretation of 'uniqueness' from the presented community data (e.g., nMDS plots or dendrograms).**

Additional studies from the turbidite sites discussed previously attempt to shed light on what characteristics of the turbidites may be driving these diversity patterns. Glover et al. (2001) compared diversity of polychaetes and other macrofauna in the NE Atlantic and Lambshead et al. (2001) reviewed meiofaunal nematode diversity at disturbed sites in the NE Atlantic and Venezuela Basin. The Madeira Abyssal Plain turbidite site was characterised by low polychaete and nematode species diversity and high dominance (Glover et al., 2001; Lambshead et al., 2001), a pattern also observed for nematodes at the

HEBBLE site, an area of the deep sea that is regularly disturbed by benthic storms (Aller, 1997; Lambshead et al., 2001). The authors of these studies interpreted the polychaete and nematode communities as being characteristic of a site that has been recolonized after a disturbance (Glover et al., 2001; Lambshead et al., 2001). Similar high dominance by various meiofauna species described as 'opportunistic early colonizers' have been noted at the Congo Channel (Van Gaever et al., 2009), Cap Breton Canyon (Hess et al., 2005; Hess and Jorissen, 2009), and off the coast of Japan (Tsujimoto et al., 2020).

However, nematode diversity at the Venezuela Basin showed no impact from the turbidite (Lambshead et al., 2001). This





difference in results between the Madeira Abyssal Plain and Venezuela Basin nematodes was attributed to the difference in the age of the respective turbidites (Madeira Abyssal Plain: 930 years, Venezuela Basin: 2,000 years) and the difference in local sedimentation rates (Madeira Abyssal Plain: 0.1-1.0 cm kyr-1 (Weaver and Rothwell, 1987), Venezuela Basin: 7.2 cm kyr$^{-2}$ (Cole et al., 1985)). The Venezuela Basin nematode community has had longer to recover, and a more pelagic-

influenced sediment regime has developed on top of the turbidite. Deep-sea macrofaunal diversity has been positively related to sediment diversity (Etter and Grassle, 1992), and similar direct links between nematode diversity and sediment characteristics have also been observed (Carman et al., 1987; Leduc et al., 2012; Tietjen, 1984). Therefore, Lambshead et al. (2001) postulated that on a local level alterations of the physical characteristics (grain size and shear strength) of the sediments (Huggett, 1987) caused by turbidites continue to affect macrofaunal and meiofaunal communities, though

meiofaunal communities appear to show a greater resilience to these disturbances than do macrofauna. However, as seen in the studies off the coast of Japan and in the Cap Breton Canyon carbon, oxygen and other chemical signals cannot be ruled out as factors that may be influencing community composition as well as community productivity (Anschutz et al., 2002; Hess et al., 2005; Hess and Jorissen, 2009; Nomaki et al., 2016; Tsujimoto et al., 2020).

**4 Conclusion and future research directions**

Evidence for the effects of turbidity flows on the productivity of faunal communities in the deep sea is ambiguous at best. From the reviewed studies, it is difficult to draw a conclusion on the general validity of Heezen et al.'s (1955) turbidity flow productivity hypothesis in the deep sea. Some studies show a positive influence on proxies of productivity, particularly biomass (Briggs et al., 1996; Carey, 1981; Griggs et al., 1969; Hess and Jorissen, 2009; Richardson et al., 1985; Thurston et al., 1994) while these same studies and others show negative or no influence on other proxies of productivity, specifically

abundance (Briggs et al., 1996; Carey, 1981; Van Gaever et al., 2009; Galéron et al., 2009; Glover et al., 2001; Gooday, 1996; Kitahashi et al., 2014, 2016; Lambshead et al., 1995, 2001; Pearcy et al., 1982; Richardson et al., 1985; Woods and Tietjen, 1985). The difficulty in distinguishing a clear effect of turbidity flows on deep-sea benthic productivity among these studies is most likely related to the particular nature of the sites used to evaluate the hypothesis (e.g., location relative to turbidity flow path, whether the impact was mostly erosional or depositional, and the time since impact by a turbidity flow)

(Fig. 1b), and possibly the type of measurement used to evaluate the impact. In contrast, the evidence in support of Angel and Rice's (1996) diversity hypothesis is relatively clear. However, even in the case of Angel and Rice's (1996) hypothesis, where evidence for lower local diversity (Briggs et al., 1996; Frutos and Sorbe, 2017; Van Gaever et al., 2009; Glover et al., 2001; Gooday, 1996; Hess and Jorissen, 2009; Kitahashi et al., 2016; Lambshead et al., 2001; Olu et al., 2017; Tietjen, 1984) and higher regional diversity (Briggs et al., 1996; Frutos and Sorbe, 2017; Van Gaever et al., 2009; Glover et al., 2001; Hess

and Jorissen, 2009; Olu et al., 2017; Tietjen, 1984) seems to exist, the driving factors for these patterns are unclear. This review has indicated that turbidity flows may be influencing benthic communities via increased carbon availability, or due to physio-chemical characteristics of the sediments, or some combination thereof. Attempting to understand how turbidity



flows impact the deep sea by looking at ancient turbidites is confounded by other natural processes, such as the flux of carbon from the surface to the benthos, and the hundreds to thousands of years since the turbidity flows occurred. Further,

the hypotheses considered here focus on the distal deposition environment, usually abyssal habitats (the environment that Heezen et al. (1955) originally specified) but turbidity flows and studies of them evaluate the impact to the benthic communities all along the flow's path. In Figure 1b and 1c we propose potential along path patterns in proxies of productivity and diversity for a large, generalized turbidity flow in a canyon at three different time points following the event.

A better way to test Heezen et al.'s (1955) hypothesis of productivity and Angel & Rice's (1996) hypothesis of regional diversity, and to understand why these impacts are occurring and in which environments, is to look at more recent turbidity flows such as those triggered by the 1999 storm in Cap Breton Canyon, France, periodic turbidity flows induced by river flooding in the Congo Channel, SE Atlantic, and the 2011 Tōhoku Earthquake off Japan. However, even these studies lack sufficient pre-disturbance data or the spatial spread of data to interpret the impacts of the turbidity flow on the benthic

community. A recent turbidity flow event in Kaikōura Canyon, New Zealand, triggered by the 2016 Kaikōura earthquake (Mountjoy et al. 2018) may provide an ideal opportunity to test turbidity flow hypotheses. Ten years before the turbidity flow, the canyon head to depths of 1,300 m were surveyed using photographic seabed transects, sediment cores, and grabs, yielding detailed information about epibenthic megafaunal, infaunal megabenthic invertebrate, and demersal fish communities which indicated that the canyon was a benthic biomass hotspot (DeLeo et al. 2010). Additional sediment cores

were collected six years before the turbidity flow providing further information on the macro- and meio-infaunal communities and cemented Kaikōura Canyon's notable contribution to regional biodiversity (Leduc et al., 2014, 2020). Then, surveys conducted at 10 weeks and 10 months after the turbidity flow event, collected comparable imagery, sediment cores, and grab samples after the event occurred, with the express purpose of quantifying changes in benthic community structure and sediment characteristics caused by the turbidity flow. Preliminary analysis of the video imagery collected at 10

weeks indicated that the once highly productive epifauna community was wiped out by the turbidity flow but that new chemosynthetic habitats were developing (Mountjoy et al., 2018). Further data collection at the same sites is planned, and analysis of this dataset will allow us to better understand the recovery patterns of the benthic communities in the immediate aftermath of a turbidity flow, and evaluate what factors influence the potential resilience of deep-sea ecosystems to this type of widespread and reoccurring disturbance.


**Author Contributions.** KTB conducted the literature review, wrote the first draft of the manuscript, and constructed the figures and table. AAR, DL, and DAB contributed to the manuscript drafting plan and figure and table design, wrote portions of the text, and edited drafts of the manuscript. All authors edited and approved the final manuscript text.

**Competing interests.** The authors declare that they have no conflict of interest.



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
