# Peer review of "Review and syntheses: Turbidity flows – evidence for effects on deep-sea benthic community productivity is ambiguous but the influence on diversity is clearer"

_Biogeosciences, 2020_

## Referee Comment (RC1) · Anonymous Referee #1 · 16 Nov 2020

The manuscript reviews the literature on seven turbidity flows, from millennial submarine landslides to contemporary mass wasting events, with the aim of establishing response patterns of the benthic fauna to such large disturbances. Two hypotheses are tested, i) that turbidity flows enhance standing stocks, and ii) that turbidity flows reduced local diversity while increasing regional diversity. Overall the manuscript is clear, informative and well-written and my comments are mainly minor. In the Introduction, Figure 1 provides a conceptual model for a turbidity flow and its consequences on the benthic fauna, Figure 2 summarizes the temporal and spatial scales of physical disturbances of different origins in the marine environment. These Figures are valuable but the rationales behind the Figures are not explained. Figure 1 is not even referenced in the Introduction and quickly mentioned in the discussion. In Figure 1, the response of the benthic community is hypothesized to vary according to the location along the flow path and according to the timing from initial disturbance but the presentation of results that follows is not structured according to these hypotheses. This Figure 1 might be better moved to the conclusion as an outcome of the review underlying the factors that may influence the response of benthic communities and the absence of clear patterns if turbidity flows are considered without specifying the erotional/depositional context nor the history/frequency of the flows. In Figure 2, because seabed mining in the deep sea rises a number of concerns, including the potential impacts of mining plumes, it might be interesting to develop the justifications for plotting seabed mining with a rather low frequency and extent of disturbances. According to the literature, a plume of sediments due to nodule mining might have an impact over hundreds of $km^2$ for daily operations lasting for decades (e.g. Glover et al. 2001). l.248-249 "there is evidence that the biomass but not the abundance of all faunal size classes is higher in turbidites than in nearby pelagic sediments" But there is also evidence that biomass was not significantly different between turbidite and pelagic (see l.239). To avoid any bias in interpretation I would suggest to delete this sentence. l.335 It might be interesting to underline here that carbon content, oxygen and ammonium are related to early diagenetic processes along a gradient of redox conditions so all of these measurements provide proxies for OM degradation. Figue 4 It is not clear how contradictory patterns in abundance and biomass were classified in the "Both" graph, some details might be needed. For example small-size opportunists may proliferate with no influence on biomass, how this will fit into Increase, Decrease or No Change? l.397-399 This is an important conclusion of this review, which is supporting the conceptual models in Figure 1, and would deserve to be elaborated here. l.406-407 Carbon availability and chemical characteristics, if they refer to oxygen and ammonium, are likely related to the same processes of OM degradation. Along a gradient of redox conditions, OM degradation first consumes oxygen, then nitrates producing ammonium, then sulfates producing hydrogen sulphides. Along this gradient, when OM inputs are large enough to deplete oxygen and nitrate, abundance and biomass increase, when OM inputs are large enough to deplete oxygen, nitrate, and sulfate then the combined effects of hypoxia and hydrogen sulphide toxicity reduce abundance and biomass, unless the detritic-based system turns into a chemosynthetic-based system.

---

## Referee Comment (RC2) · Anonymous Referee #2 · 22 Dec 2020

What a great paper, and a review that is probably long overdue. It is extremely well written and nicely illustrated. In fact it is so well written I have no real detailed comments to make.

I think title feels a bit like a holding statement until a proper title is decided upon. For example "evidence for effects on deep sea benthic community productivity is ambiguous" what does that mean? This is more ambiguous. Surely the collapse of the seafloor has to be a negative effect to those on said seafloor and those that the seafloor is about to pile down on? Then, again ambiguously "the influence on diversity is clearer"- is that

a positive influence or a negative one? It reads like 'Turbidity flows probably don't do much for productivity but might do something to diversity.' I think a more succinct title will do the paper greater justice.

It might also be worth mentioning the significance of post-turbidity flow surveys to other contemporary research. I am thinking deep sea mining. There has been a lot of talk recently about burial and smothering in the wake of polymetallic nodule field disturbance, and what that secondary disturbance might do to the benthic fauna. This could be incorporated into the discussion to give a more applied value to this review.

---

## Author Comment (AC1) · 26 Jan 2021

Reviewer 1 Comments
*Figure 1 is not even referenced in the Introduction and quickly mentioned in the discussion.*

The non-reference to this figure was an error having been left out from an earlier draft. This error has been rectified and the figure is now referenced in the Introduction and Section 2 (l.28 and l.142-143). Additionally, an adjustment has been made to where it was originally referenced in Section 3 (l.358).

*In Figure 1, the response of the benthic community is hypothesized to vary according to the location along the flow path and according to the timing from initial disturbance but the presentation of results that follows is not structured according to these hypotheses.*

Figure 1 is intended as a summary of the forces along a hypothetical turbidity flow path, and how these interplay with response patterns in benthic productivity and diversity. We have structured our review on two previously published hypotheses, for productivity and diversity, and have used the figure (in addition to use in the Introduction) to help illustrate some of the potential controlling forces that relate to these hypotheses in Sections 2 and 3.

*This Figure 1 might be better moved to the conclusion as an outcome of the review underlying the factors that may influence the response of benthic communities and the absence of clear patterns if turbidity flows are considered without specifying the erotional/depositional context nor the history/frequency of the flows.*

We agree with the reviewer that the figure should be referenced in the Conclusion (in addition to earlier sections), as a reminder for the reader that one of the outputs of this review is a useful summary of turbidity flow structure and physical processes, and their influences of turbidity flows on benthic communities. References to Figure 1 are now throughout the text (l.28, l.143-144, l.358, l.410, l.411, and l.421).

*In Figure 2, because seabed mining in the deep sea rises a number of concerns, including the potential impacts of mining plumes, it might be interesting to develop the justifications for plotting seabed mining with a rather low frequency and extent of disturbances. According to the literature, a plume of sediments due to nodule mining might have an impact over hundreds of km2 for daily operations lasting for decades (e.g. Glover et al. 2001).*

We only included the direct disturbance of mining (reason for the small ellipse) in the figure, but the reviewer is correct that we should have also included the potential disturbance from the mining plume. We have now updated the ellipse for seabed mining in Figure 2.

*l.248-249 "there is evidence that the biomass but not the abundance of all faunal size classes is higher in turbidites than in nearby pelagic sediments" But there is also evidence that biomass was not significantly different between turbidite and pelagic (see l.239). To avoid any bias in interpretation I would suggest to delete this sentence.*

We have amended the sentence to clarify our point that there is "some" evidence, i.e., the evidence is equivocal (l.266).

*l.335 It might be interesting to underline here that carbon content, oxygen and ammonium are related to early diagenetic processes along a gradient of redox conditions so all of these measurements provide proxies for OM degradation.*

We agree, and have added a sentence to acknowledge the process noted by the reviewer (l.332-334).

*Figue 4 It is not clear how contradictory patterns in abundance and biomass were classified in the "Both" graph, some details might be needed. For example small-size opportunists may proliferate with no influence on biomass, how this will fit into Increase, Decrease or No Change?*

On consideration of this comment, it is evident that it is not necessary to include the 'Both' category (which is just a sum of the other two categories), so we have removed it and modified the figure and caption accordingly.

*l.397-399 This is an important conclusion of this review, which is supporting the conceptual models in Figure 1, and would deserve to be elaborated here.*

As suggested, we have elaborated on this point (l.399-404).

*l.406-407 Carbon availability and chemical characteristics, if they refer to oxygen and ammonium, are likely related to the same processes of OM degradation. Along a gradient of redox conditions, OM degradation first consumes oxygen, then nitrates producing ammonium, then sulfates producing hydrogen sulphides. Along this gradient, when OM inputs are large enough to deplete oxygen and nitrate, abundance and biomass increase, when OM inputs are large enough to deplete oxygen, nitrate, and sulfate then the combined effects of hypoxia and hydrogen sulphide toxicity reduce abundance and biomass, unless the detritic-based system turns into a chemosynthetic-based system.*

We have included this process in response to an earlier comment (l.332-334), and have amended the noted sentence to include a fuller explanation (l.412).

---

## Author Comment (AC2) · 26 Jan 2021

Reviewer 2 Comments

*I think title feels a bit like a holding statement until a proper title is decided upon. For example "evidence for effects on deep sea benthic community productivity is ambiguous" what does that mean? This is more ambiguous. Surely the collapse of the seafloor has to be a negative effect to those on said seafloor and those that the seafloor is about to pile down on? Then, again ambiguously "the influence on diversity is clearer"- is that a positive influence or a negative one? It reads like 'Turbidity flows probably don't do much for productivity but might do something to diversity.' I think a more succinct title will do the paper greater justice.*

We agree with the reviewer that a succinct title is more desirable, and have now changed it to: Review and syntheses: Impacts of turbidity flows on deep-sea benthic communities.

*It might also be worth mentioning the significance of post-turbidity flow surveys to other contemporary research. I am thinking deep sea mining. There has been a lot of talk recently about burial and smothering in the wake of polymetallic nodule field disturbance, and what that secondary disturbance might do to the benthic fauna. This could be incorporated into the discussion to give a more applied value to this review.*

We agree with the reviewer, and had included some sentences to this effect in an earlier draft. We have now re-included them (l.434-435), and have added text to the abstract to highlight this point (l.21-22).